# Observational activation of anterior cingulate cortical neurons coordinates hippocampal replay in social learning

Xiang Mou[1]*, Daoyun Ji[1,2]*

[1]Department of Neuroscience, Baylor College of Medicine, Houston, United States; [2]Department of Molecular and Cellular Biology, Baylor College of Medicine, Houston, United States

## eLife Assessment

This study provides **convincing** evidence of coordinated spiking activity of neurons in the anterior cingulate cortex (ACC), and correlated activity in the CA1 subregion of the hippocampus, during observational learning. The authors also show coordinated ACC-CA1 neural activity during rest periods prior to the performance of the observationally learned task. The **important** findings significantly advance the field's understanding of neural mechanisms underlying social learning.

*For correspondence:
xmou@bcm.edu (XM);
dji@bcm.edu (DJ)

**Competing interest:** The authors declare that no competing interests exist.

**Abstract** Social learning enables a subject to make decisions by observing the actions of another. How neural circuits acquire relevant information during observation to guide subsequent behavior is unknown. Utilizing an observational spatial working memory task, we show that neurons in the rat anterior cingulate cortex (ACC) associated with spatial trajectories during self-running in a maze are reactivated when observing another rat running the same maze. The observation-induced ACC activities are reduced in error trials and are correlated with activities of hippocampal place cells representing the same trajectories. The ACC activities during observation also predict subsequent hippocampal place cell activities during sharp-wave ripples and spatial contents of hippocampal replay prior to self-running. The results support that ACC neurons involved in decisions during self-running are reactivated during observation and interact with hippocampal replay to guide subsequent spatial navigation.

## Introduction

Observational learning, a form of social learning, takes the form of watching and/or imitating the behavior of others to learn (*Bandura, 1997*). This type of learning is widespread in a range of species, including humans and rodents (*Heyes and Galef, 1996*; *Meltzoff et al., 2009*). Unlike learning through self-action, observational learning often involves observers making decisions contingent upon others' actions. Although recent studies have begun to reveal the neural activities associated with observational learning (*Allsop et al., 2018*; *Danjo et al., 2018*; *Jeon et al., 2010*; *Leggio et al., 2000*; *Mou and Ji, 2016*; *Olsson and Phelps, 2007*; *Omer et al., 2018*), it remains unknown how neural circuits process specific information acquired during observation to guide subsequent behavior.

The anterior cingulate cortex (ACC), a brain area involved in decision-making (*Kane et al., 2022*), also plays a key role in observational learning (*Burgos-Robles et al., 2019*). On one hand, previous studies show that ACC neurons are modulated by emotional attributes important for decision-making, such as reward or shocks associated with stimuli or places (*Johansen et al., 2001*; *Mashhoori et al., 2018*). Strong evidence also exists that activities of ACC neurons encode the valence or value of stimuli

to be used for decision-making or as a result of decision-making (*Caracheo et al., 2018*; *Hillman and Bilkey, 2010*; *Monosov, 2017*). On the other hand, ACC can be activated when observing another subject in pain or receiving a reward (*Carrillo et al., 2019*; *Schneider et al., 2020*). ACC is also required for an observer to acquire fear responses to a context or stimulus after observing another subject doing so (*Jeon et al., 2010*; *Olsson and Phelps, 2007*), and ACC neurons are activated during fear acquisition through observation (*Allsop et al., 2018*).

These known functions of ACC raise an important question: How are specific ACC neuronal activities occurring during observation involved in the observer's subsequent decisions? To gain insights into this question, we set out to study ACC neurons in rats performing an observational spatial working memory task (oSWM) (*Mou et al., 2022*), during which an observer rat (OB), after observing the run of a demonstrator (Demo) in the maze, decides which spatial trajectories in a T-maze to run for water reward, from a nearby, separate observation box.

Since places and spatial trajectories are encoded by hippocampal place cells (*O'Keefe and Dostrovsky, 1971*; *Wilson and McNaughton, 1993*), it is not surprising that the oSWM task depends on the CA1 area of the hippocampus (*Mou et al., 2022*). It is known that, as an animal travels through a spatial trajectory, hippocampal place cells are activated sequentially (*Harris et al., 2003*). The sequential activation can be replayed during awake resting or reward consumption (*Diba and Buzsáki, 2007*; *Foster and Wilson, 2006*), together with the concurrent high-frequency sharp-wave ripples (SWRs) in the CA1 local field potentials (LFPs) (*Buzsáki et al., 1992*; *Csicsvari et al., 2000*). Evidence exists that such awake replay may be a neural substrate underlying memory recall or planning for future spatial trajectories (*Carr et al., 2011*; *Jadhav et al., 2012*; *Pfeiffer and Foster, 2013*; *Wu et al., 2017*). Indeed, in the oSWM task, awake replay of CA1 place cell activities associated with self-running in the T-maze occurs remotely in the observation box, and this remote awake replay predicts subsequent self-running trajectories in the maze (*Mou et al., 2022*), consistent with a role of awake replay in spatial planning.

In this study, we examine ACC neurons in the oSWM task and their interactions with CA1 place cells, aiming to understand how ACC activities during observation are related to the hippocampal replay that presumably plans for future spatial trajectories. We focus on the hypothesis that ACC neurons acquire the information key to spatial decisions during observation and use it to coordinate the replay of CA1 place cells for planning subsequent spatial trajectories during self-running.

## Results

The data analyzed in this study were obtained from 16 rats, each as a well-trained OB performing the oSWM task. Briefly, in each trial of the task, an OB stayed in an observation box while a Demo ran along a nearby T-maze and made a random choice of a left or right (outbound) trajectory for water (*Figure 1A*). The OB was rewarded with water in the box if the animal poked on the same side of the box as the Demo's in the maze within 10 s. After the Demo returned along a left or right (inbound) trajectory and was then confined in a rest box, the OB was transported to the central arm of the T-maze. The OB was rewarded with water if the animal chose the same outbound trajectory as the Demo's. The OB then returned along the same inbound trajectory in the maze as the Demo's, before being moved back to the observation box for the next trial. Each daily session consisted of ~40 trials. After training, all OBs learned to synchronize their pokes in the box with their Demo's pokes in the maze and followed their Demo's trajectories during subsequent self-running in ~85% of the trials (*Mou et al., 2022*).

### ACC dependence of the oSWM task

Before analyzing ACC neurons, we first asked whether the OBs' performance required an intact ACC. We infused *N*-methyl-D-aspartate (NMDA) into the ACC to induce lesions in a group of OBs (*N*=5) and compared their performance to another group (*N*=5) with vehicle infusion (*Figure 1B*) in three sessions tested 2 weeks after (After) and in three sessions before (Before) the infusion.

We computed a poke performance curve for each session to measure the synchronization of pokes between an OB in the box and the corresponding Demo in the maze (see Methods) (*Mou et al., 2022*). The average poke performance curve of the vehicle group in After peaked close to time 0, whereas that of the NMDA group displayed a delayed, weak peak (*Figure 1C*). The curves were

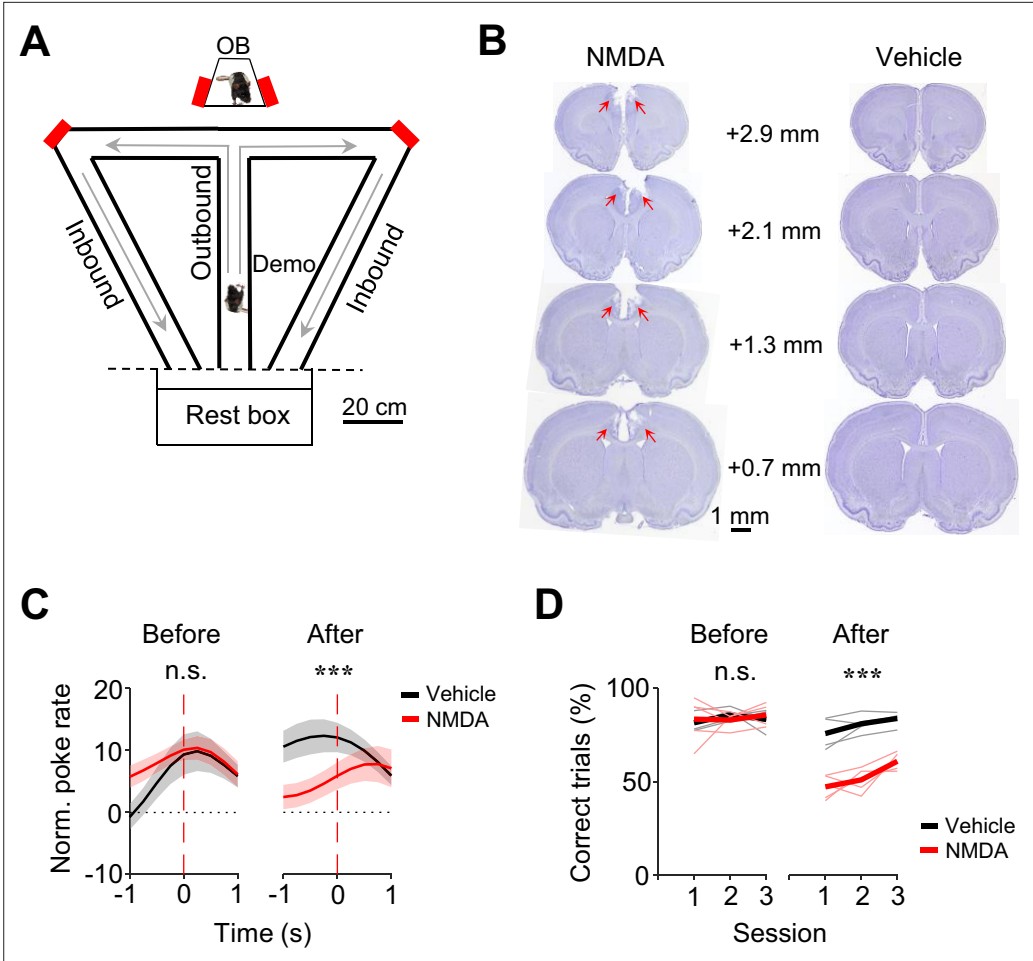

**Figure 1.** OB performances in the observation box and in the maze depended on anterior cingulate cortex (ACC). (**A**) Behavioral apparatus consisting of an observation box (top), a continuous T-maze and a rest box (bottom). Red: reward site (water port); arrow: running direction. OB/Demo: observer/demonstrator rat. (**B**) Coronal brain sections after *N*-methyl-D-aspartate (NMDA) and vehicle infusion. Arrow: lesion in ACC. Number: anteroposterior coordinate from the Bregma. (**C**) Average (mean ± SEM) poke performance curves of OBs in the box for the three sessions before (Before) and the first three testing sessions 2 weeks after NMDA or vehicle infusion (After). Time 0: OB's poke time. (**D**) Maze performances of OBs in each of Before and After sessions. Thin/thick line: individual rat/ average performance. ***: p<0.001, **: p<0.01, *: p<0.05, n.s.: not significant (same for all figures).

significantly different between the two groups in After (*two-way ANOVA: $F_{(1,329)} = 19$, p=1.7 × $10^{-5}$), but not in Before ($F_{(1,329)} = 3.6$, p=0.060). This difference was mainly due to the decreased performance in the NMDA group after infusion (Before vs. After, *two-way ANOVA: $F_{(1,329)} = 6.4$, p=0.012). Therefore, the performance of the NMDA group in the box was impaired by the ACC lesion.

In the maze, the performance of the NMDA group, measured by the percentage of correct trials, was also significantly lower than that of the vehicle group in After (NMDA: 53.3 ± 2.0%, mean ± SEM, *N*=15 sessions from 5 rats; Vehicle: 80.6 ± 1.6%, *N*=15; *two-way ANOVA: $F_{(1,29)} = 134$, p=9.2 × $10^{-12}$), but not in Before (NMDA: 84.1 ± 1.9%, *N*=15; Vehicle: 83.6 ± 1.1%, *N*=15; *two-way ANOVA: $F_{(1,29)} = 2.7$, p=0.11; *Figure 1D*). The difference in After was not due to a locomotion deficit, because there was no significant difference in the mean duration of running laps between the two groups in After (NMDA: 64.1±1.7 s, Vehicle: 62.9±1.7 s, p=0.60, two-sided *t*-test). There appeared to be an increase in the later session of After, which could be due to re-learning after the lesion. Overall, the data indicate that the ACC lesion significantly impaired the maze performance of the NMDA group. Therefore, besides the CA1 area of the hippocampus (*Mou et al., 2022*), the oSWM task also depends on ACC.

## Activation of ACC selective cells during observation

We next analyzed ACC cells in the oSWM task, which were recorded simultaneously with CA1 place cells from six well-trained OBs. In total, 492 ACC cells were obtained from OBs in 19 sessions with a well-trained Demo as described above, referred to as the standard Demo condition. In addition, we also recorded ACC cells in two control conditions (see Methods): 248 ACC cells from 3 OBs in 11 sessions with the Demo replaced by a moving object (Object) and 309 ACC cells from 5 OBs in 13 sessions with the Demo removed (Empty).

We first defined left and right trials of a session by the Demo's choice of the left and right side of the maze, respectively. Since the OBs were well trained, they also ran the same left (or right) trajectories correctly in the maze in the majority of the left (or right) trials, after getting rewarded in the left (or right) side of the box (although a well-trained OB sometimes poked on the wrong side of the box first, the rat quickly switched to the other side and got rewarded in the box in almost all the left/right trials).

We noticed that many ACC cells distinguished the left from the right side of the maze during self-running. We call such cells ACC selective cells. For example, an ACC left-selective cell fired with higher rates on the left inbound trajectory in left trials than on the corresponding trajectory in right trials, and an ACC right-selective cell had higher rates on a segment of the right outbound trajectory in right trials than on the corresponding segment in left trials (*Figure 2A*). Because previous studies showed that ACC cells encode valence, value, or decision associated with a stimulus or location (*Caracheo et al., 2018*; *Hillman and Bilkey, 2010*; *Kane et al., 2022*; *Mashhoori et al., 2018*; *Monosov, 2017*), we reasoned that the activities of such ACC selective cells likely reflected the decision or reward associated with the chosen trajectory or segment during self-running. Since our goal was to investigate the function of ACC cells in observational learning, not the exact attributes of a trajectory/segment to be encoded during self-running, we focused on whether and how these ACC cells reflecting the left or right choice in the maze were also activated during observation in the box.

To quantitatively identify ACC selective cells, we computed the firing rate of each ACC cell during each running lap on the left or right side of the maze (inbound and outbound trajectories combined, without identifying specific active segments). We assigned a selectivity index (SI) to quantify the difference in the mean rate between left and right laps (see Methods), which ranged between –1 (most left-selective) and 1 (most right-selective). An ACC cell was classified as selective if the difference was statistically significant. Indeed, the two example cells above were classified as selective cells (left-selective cell: $p=3.2 \times 10^{-3}$, two-sided $t$-test; right-selective cell: $p=2.6 \times 10^{-11}$; *Figure 2C*). A total of 202 ACC cells (41%) were selective during self-running in the T-maze under the Demo condition (under Demo). Only ACC selective cells were included in the rest of the analysis on ACC neural activities, unless specified otherwise.

We examined firing activities of these ACC cells in the box in each trial, particularly during a delay window between the time a Demo crossed the choice point in the maze and when the corresponding OB made the first correct (rewarded) poke in the box (*Figure 2—figure supplement 1A*). We focused on this delay window because it was the time for the OB to observe the Demo's choice. The distribution of this window's duration across all trials of all OBs under Demo had a dominant peak at ~2 s (*Figure 2—figure supplement 1B*). In order to examine ACC cells within a time period with relatively consistent behavior across all trials (animals with occasional long delay windows typically explored other parts of the box or showed signs of inattention before poking), we defined a delay period for each trial as the 2 s period before the OB's first correct poke in the box. The head position and head direction of a typical OB within this 2 s delay period for all left and right trials of a session displayed consistent left and right swings as expected (*Figure 2—figure supplement 1C and D*).

We found that many ACC cells fired differently during delay periods in the box between left and right trials. For example, the ACC cells selective to the left or right side of the maze (*Figure 2A*) also had higher firing rates during delay periods of left or right trials in the box (*Figure 2B*), meaning that they were also left- and right-selective in the box with significant SIs (left-selective cell: $p=4.6 \times 10^{-7}$, two-sided $t$-test; right-selective cell: $p=1.3 \times 10^{-4}$; *Figure 2C*).

We computed the SI in the maze and in the box for each ACC selective cell under Demo and found that the maze and box SIs were significantly correlated ($R=0.25$, $p=1.4 \times 10^{-4}$, *Pearson's r*; *Figure 2D*). We counted the number of ACC cells with significant SIs on the same side of the box and the maze (same-side selectivity) and compared them to the chance-level distribution, obtained by randomly shuffling each cell's rates during delay periods among left and right trials (see Methods). We found

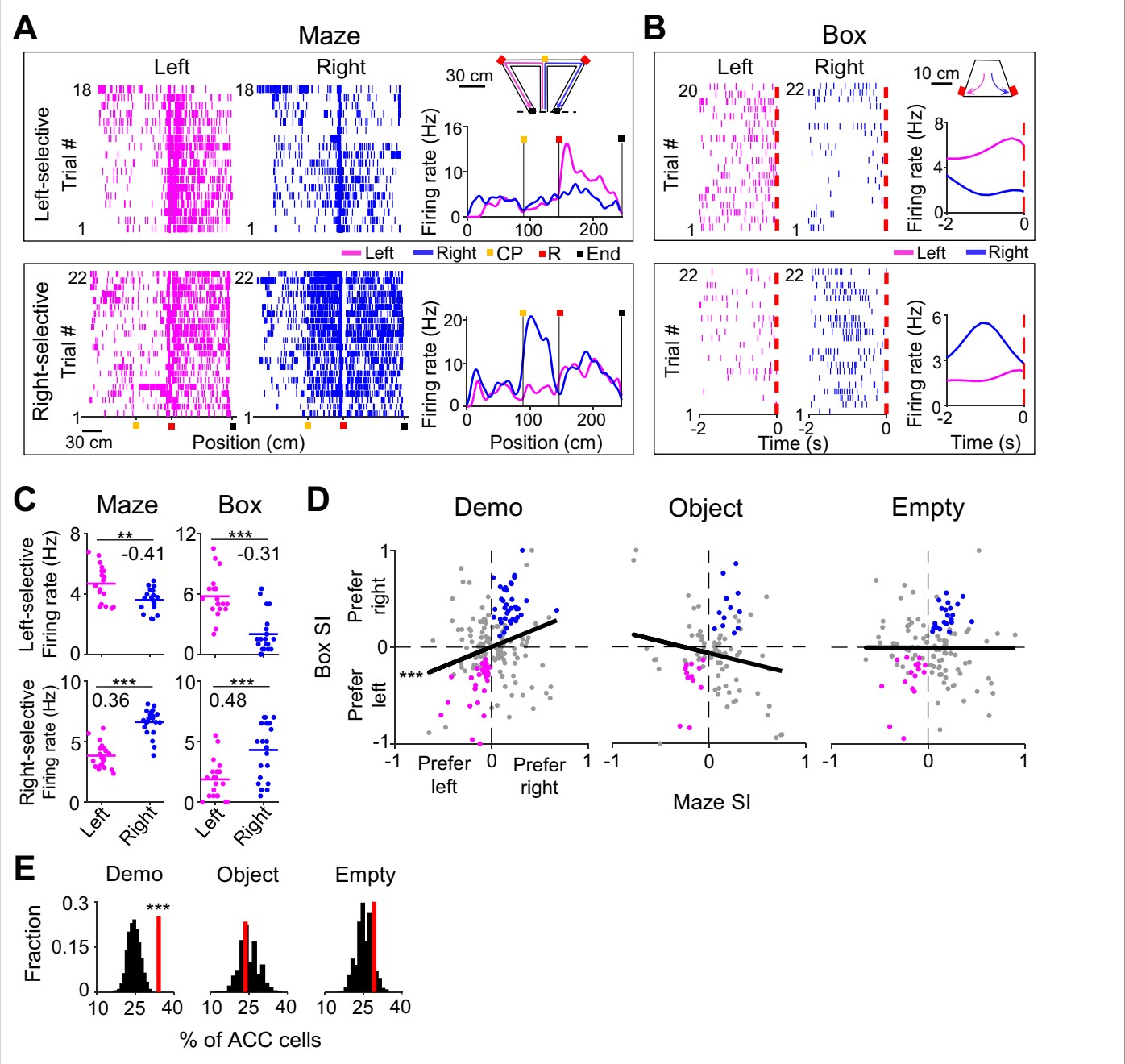

**Figure 2.** Anterior cingulate cortex (ACC) selective cells were activated during delay periods in the observation box. (**A**) Spike raster of an ACC left- (top) and an ACC right-selective (bottom) cell on the left (magenta) and right (blue) sides of the T-maze, and their average firing rates along the (linearized) trajectories on the two sides (right curves). Each tick is a spike. Each row is a trial. Color markers show maze landmarks: choice point (CP), reward site (**R**), end of side arm (End). The two cells were from two different sessions. Error trials (top: 2 left and 4 right; bottom: 0) are excluded. (**B**) Similar to (**A**), but for the same two cells during the 2 s delay period of each trial in the box (dashed line: time 0 – first rewarded poke in the box). Note the higher rate on the same side of each cell's selectivity. (**C**) Trial-by-trial firing rates of the same two cells in (**A–B**) during running on the left/right side of the maze and during the delay period on the left/right side of the box. Number: selection index (SI). Note the similar SIs in the maze and in the box. (**D**) SIs in the maze and in the box for all ACC selective cells under Demo, Object, and Empty. Each dot is a cell. Black line: linear regression between maze and box SIs. Colored dots: cells with significant same-side selectivity in the box on the left (magenta) or right (blue). (**E**) Percentages of ACC cells with significant same-side selectivity in the box (left/right combined, red lines) under Demo, Object, and Empty, compared to random distributions (black) obtained by shuffling each cell's firing rates among all delay periods of left/right trials.

The online version of this article includes the following figure supplement(s) for figure 2:

**Figure supplement 1.** Behavior of observer rats (OBs) during delay periods in the observation box.

**Figure supplement 2.** Anterior cingulate cortex (ACC) cells with opposite-side selectivity in the box were fewer than the chance level under Demo.

that 34% of ACC selective cells had same-side selectivity, which was significantly higher than the chance level ($Z$=4.1, p=2.1 × 10$^{-5}$, Z-test, *Figure 2E*).

We performed the same analysis on ACC cells recorded in sessions under the two control (Object, Empty) conditions. Under the Object condition, the Demo was replaced by a moving object, which mimicked the movement of a Demo in the T-maze (see Methods). A trial was considered correct if the OB's choice in the maze was the same water port the object activated. Under the Empty condition, the Demo was removed and the T-maze was left empty. The water ports were manually triggered by a wood pole (see Methods). The number of correct trials was close to chance level (50%) under both Object and Empty conditions, which still allowed for the identification of ACC selective cells. The percentage of ACC selective cells (among all recorded ACC cells) identified under these conditions was comparable to that under Demo (*Figure 2—figure supplement 2A*). However, their SIs in the maze were not correlated with those in the box (Object: $R$=–0.19, p=0.97; Empty: $R$=–0.0019, p=0.51, *Pearson's r*; *Figure 2D*). In addition, the number of ACC cells with same-side selectivity was not significantly different from the chance level under either Object (24%, $Z$=–0.47, p=0.68, Z-test) or Empty (28%, $Z$=1.1, p=0.14; *Figure 2E*).

As a control, we also analyzed the ACC cells with opposite-side selectivity in the box. We found that the number of these ACC cells was significantly less than the chance level under Demo and not significantly different from the chance level under Object or Empty (*Figure 2—figure supplement 2B and C*).

Taken together, our data show that ACC cells selective to the left/right side of the maze were also selective to the same side in the box during delay periods under Demo condition. The presence of same-side selectivity of ACC cells, in contrast to opposite-side selectivity, indicates that the ACC cells selective to the observed trajectories in the maze were preferentially activated during observation in the box, prior to self-running on the same trajectories. The lack of same-side selectivity of ACC cells under Object or Empty conditions suggests that this activation was not simply due to the movement of OBs to the same side in the box. Therefore, our data support that observation activates those ACC cells specifically associated with the observed, and thus the subsequent future, trajectories.

## Reduced ACC activation during observation in error trials

To provide further evidence for the involvement of ACC activation in subsequent self-running, we asked whether the activation during observation was compromised in error trials. An error trial was defined as one in which an OB ran to the side in the maze opposite to the Demo's choice, even after receiving water in the box on the same side (*Figure 3A and B*). Since the OBs in this study were well trained, error trials were rare under Demo (11 ± 2% per session). Nevertheless, we examined ACC cells in those sessions that contained at least three error trials.

For an ACC cell with same-side selectivity, we compared its activities during delay periods between correct and error (≥3) trials on its preferred side of the box (left for ACC left-selective cells, right for ACC right-selective cells). ACC cells typically displayed higher rates in correct trials than in error trials under Demo, which was quantified by a positive rate difference index (DI; *Figure 3A and B*). For those ACC cells with same-side selectivity, their mean DI was significantly different from 0 under Demo (0.13±0.04, $N$=52, p=0.0016, two-sided t-test), but not under Object (–0.03±0.02, $N$=82, p=0.20) or Empty (0.01±0.02, $N$=127, p=0.51; *Figure 3C*). The mean DI under Demo was also significantly higher than those under Object and Empty ($F_{(1,257)}$ = 9.3, p=1.2 × 10$^{-4}$, *one-way ANOVA*). In addition, the fraction of ACC cells with same-side selectivity that displayed a significant positive DI was greater than the chance level, obtained by randomizing rates among correct and error trials, under Demo (14%, $Z$=3.5, p=2.1 × 10$^{-4}$, Z-test), but not under Object (3.8%, $Z$=–0.62, p=0.73) or Empty (3.9%, $Z$=–0.46, p=0.68; *Figure 3D*). For ACC cells with same-side selectivity, their mean DI during delay periods on their non-preferred side (left for ACC right-selective cells, right for ACC left-selective cells) was not significantly different from 0, and the fraction of cells with a significant positive DI was no more than the chance level under Demo, Object, or Empty (*Figure 3—figure supplement 1*).

Our data thus show that the activation of ACC cells with same-side selectivity during observation was reduced when OBs made errors during self-running in the maze, consistent with a role of observation-activated ACC activities in subsequent spatial decisions.

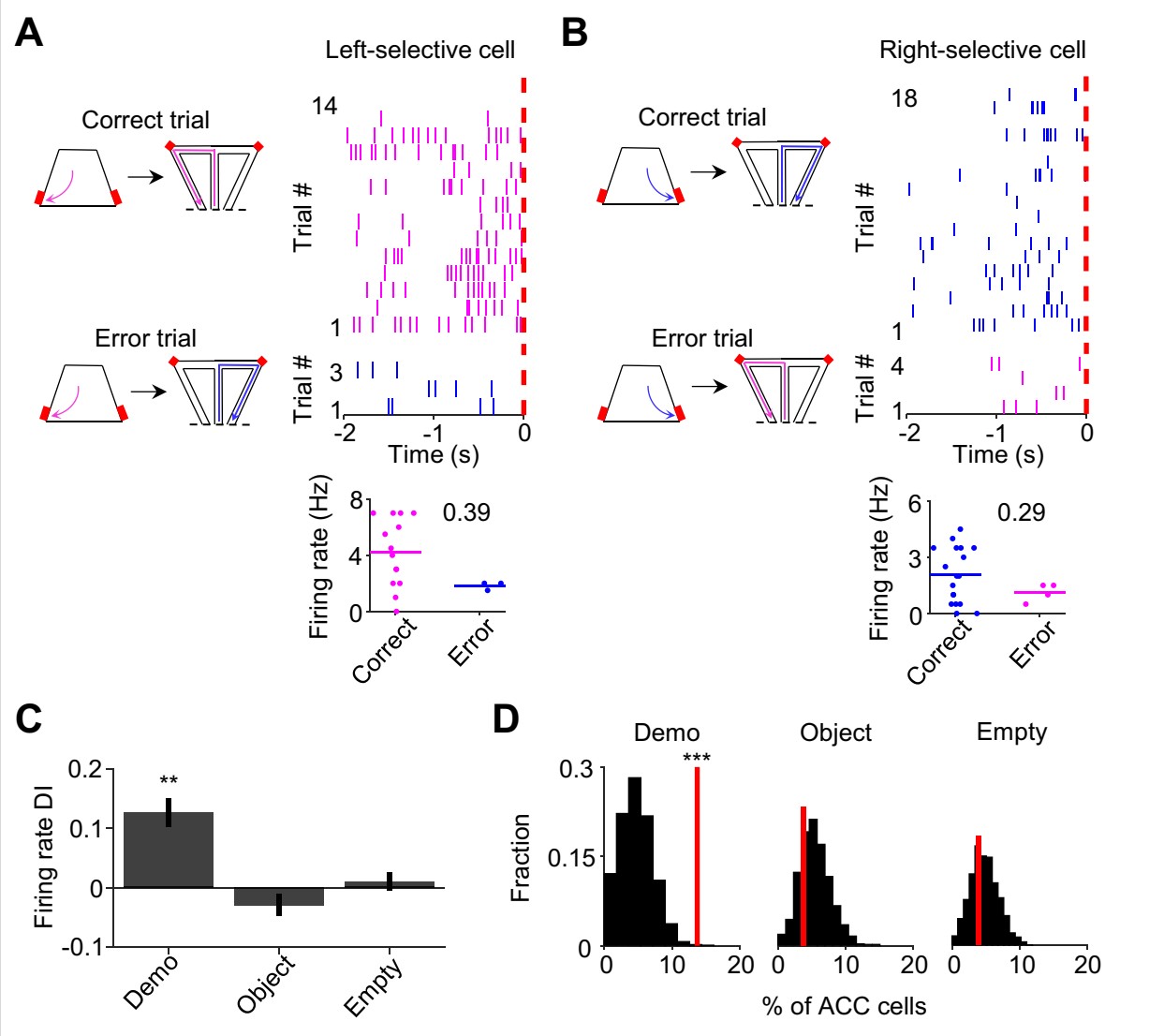

**Figure 3.** Anterior cingulate cortex (ACC) cells with same-side selectivity in the box were less activated during delay periods of error trials. (**A**) Left: illustration of a correct and an error left trial (the Demo chose left in the maze). A trial was considered an error if the observer rat (OB) chose the opposite (right) side in the maze, after getting rewarded (on the left side) in the box. Red: reward site. Right: trial-by-trial spike raster (top) of an ACC left-selective cell (in both the maze and box) and firing rates of this ACC cell in each correct/error trial (bottom) during delay periods of an example session. Each tick represents a spike. Magenta/blue: spikes in correct/error trials. Red dash line: OB's poke time (time 0). Number: firing rate difference index (DI) between correct and error trials for this cell. (**B**) Similar to (**A**), but for an ACC right-selective cell in correct and error right trials. Blue/magenta: correct/error trials. (**C**) Comparison of ACC cell DIs described in (**A–B**) during delay periods on their preferred side under Demo, Object, and Empty. (**D**) Percentages of ACC cells with same-side selectivity in the box that had significantly higher firing rate during delay periods of correct than error trials on their preferred side (left-selective and right-selective cells combined, red lines) under Demo, Object, and Empty, compared to their shuffle-generated distributions (black).

The online version of this article includes the following figure supplement(s) for figure 3:

**Figure supplement 1.** Anterior cingulate cortex (ACC) cells with same-side selectivity in the box did not differ in firing rate during delay periods between correct and error trials on their non-preferred side.

## Interaction between ACC cells and CA1 place cells during observation

Since spatial trajectories in the maze are encoded by hippocampal place cells (*Harris et al., 2003*; *O'Keefe and Dostrovsky, 1971*; *Wilson and McNaughton, 1993*), we next asked whether ACC cells selective to a trajectory in the maze interacted with those CA1 place cells encoding the same trajectory during delay periods in the box.

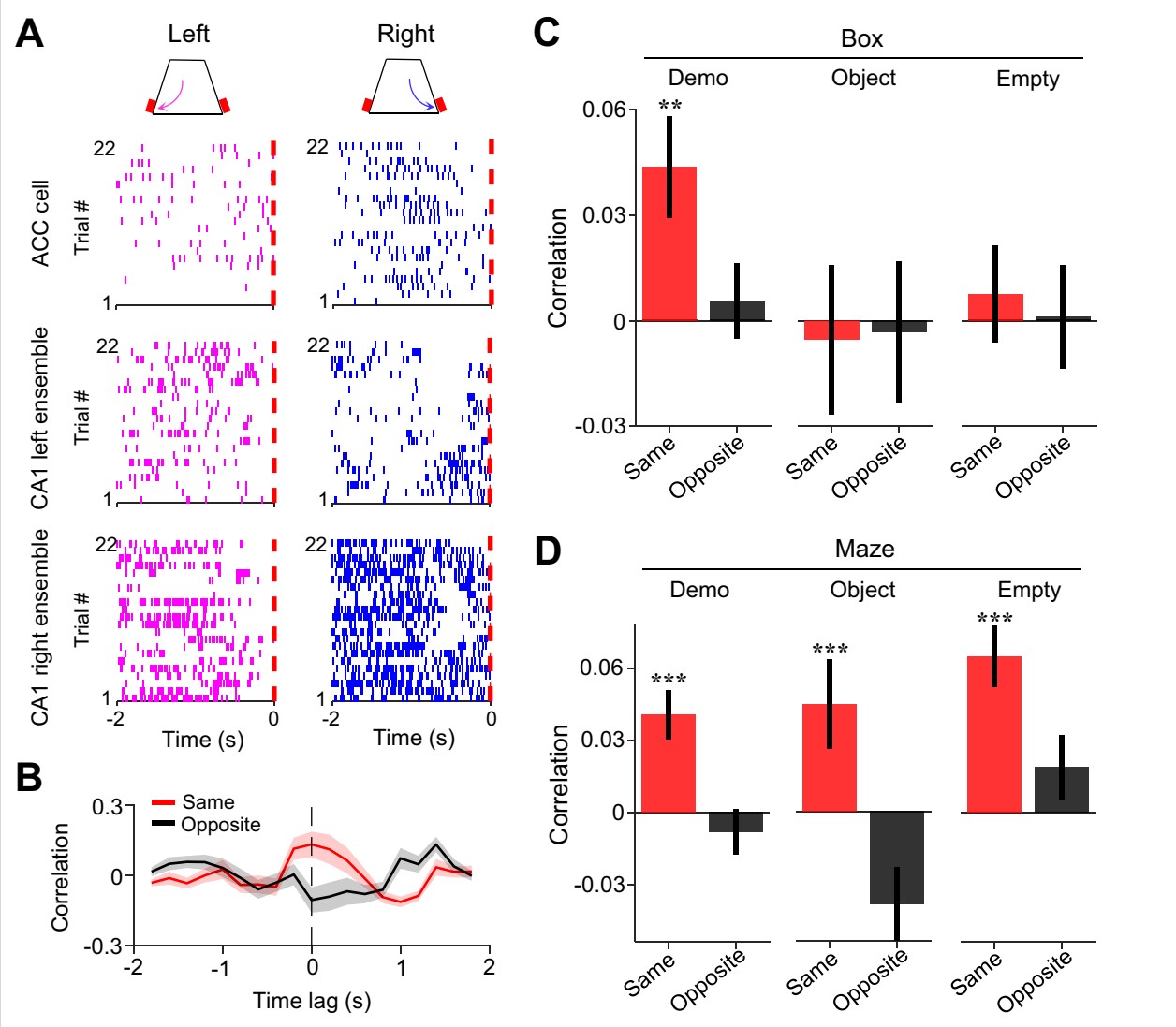

**Figure 4.** Activities of anterior cingulate cortex (ACC) cells were correlated with those of CA1 same-side ensembles during delay periods in the box. (**A**) Trial-by-trial spike raster of an example ACC right-selective cell and its associated CA1 left and right ensembles during delay periods of left (magenta) and right (blue) trials of a session. Each tick is a spike. Red dash line: time of first rewarded poke in the box (time 0). (**B**) Firing rate cross-correlation between the example ACC cell and its associated CA1 ensembles in (**A**). Red/black curves: average values over all trials in the session. Shaded areas: SEM. Note the peak correlation around time lag 0 (dashed line) between the ACC cell and its same-side CA1 ensemble. (**C**) Average correlations (mean ± SEM, at time lag 0) during delay periods between ACC cells and their CA1 same- (red) and opposite-side (black) ensembles under Demo, Object, and Empty. Note the significant same-side correlation under Demo, but not under Object or Empty. (**D**) Similar to (**C**), but for correlations during running laps in the maze. Note the significant same-side correlation under all three conditions, as expected.

The online version of this article includes the following figure supplement(s) for figure 4:

**Figure supplement 1.** CA1 cells with same-side selectivity in the box were not significantly different from the chance level.

We identified the CA1 place cells active during self-running in the maze. A total of 762, 499, and 621 CA1 place cells were identified under Demo, Object, and Empty, respectively. We checked whether CA1 place cells selectively active on left/right trajectories in the maze displayed similar selectivity during delay periods in the box and found no significant same-side or opposite-side selectivity under either Demo, Object, or Empty (*Figure 4—figure supplement 1*). Since CA1 cells often had relatively low mean rates (<2 Hz) during delay periods (*Figure 4—figure supplement 1A*), we examined CA1 place cells at the population level by combining those selective to the left or right trajectories of the maze (inbound and outbound combined) in each session into a left or right CA1 ensemble (see Methods). We then asked how ACC cells selective to the left/right side of the maze interacted

with the CA1 ensembles selective to the same or opposite side (same- or opposite-side ensembles) during delay periods in the box (*Figure 4A*).

For this purpose, we computed a cross-correlation curve between the firing rates of an ACC cell and those of a CA1 ensemble (all spikes of all cells in an ensemble combined) within 200 ms time bins of delay periods. As shown in *Figure 4B*, an example ACC cell displayed a cross-correlation curve with a peak around the time lag 0 with its same-side CA1 ensemble, but not with its opposite-side one. We thus used the correlation value at the time lag 0 to quantify the interaction between an ACC cell and a CA1 ensemble.

During delay periods in the observation box under Demo (*Figure 4C*), ACC cells on average had a significant mean correlation with their CA1 same-side ensembles (same-side correlation: 0.044±0.014, $N=202$, p=0.0028, two-sided $t$-test compared to 0), but not with their opposite-side ones (opposite-side correlation: 0.0056±0.011, $N=202$, p=0.60). In contrast, there was no significant same- or opposite-side correlation under either Object (same-side: –0.0053±0.021, $N=101$, p=0.80; opposite-side: –0.0031±0.020, $N=101$, p=0.88) or Empty (same-side: 0.0075±0.014, $N=130$, p=0.59; opposite-side: 0.0012±0.015, $N=130$, p=0.94). The same-side correlation was significantly higher than the opposite-side one under Demo (p=0.0020, two-sided *paired t*-test), but not under Object (p=0.89) or Empty (p=0.62; *Figure 4C*).

To verify the validity of this method, we performed the same analysis on the activities of ACC cells and CA1 ensembles during self-running in the maze (*Figure 4D*). The mean same-side correlation was significant under all conditions (Demo: 0.041±0.010, $N=202$, p=4.9 × 10⁻⁵, two-sided $t$-test compared to 0; Object: 0.045±0.019, $N=101$, p=0.0091; Empty: 0.065±0.013, $N=130$, p=7.1 × 10⁻⁷), but not the opposite-side correlation (Demo: –0.0081±0.0094, $N=202$, p=0.80; Object: –0.039±0.015, $N=101$, p=0.99; Empty: 0.019±0.013, $N=130$, p=0.084). The same-side correlations were also significantly higher than the opposite-side ones under all conditions (Demo: p=2.2 × 10⁻¹⁴, two-sided *paired t*-test; Object: p=9.1 × 10⁻¹⁰; Empty: p=2.5 × 10⁻⁷). Because ACC selective cells and their same-side CA1 ensembles were activated together during self-running under all conditions, this result shows that the method correctly identified the expected ACC-CA1 interactions during self-running in the maze.

Our data thus show that ACC selective cells were temporarily correlated with CA1 place cells encoding the same maze trajectories during delay periods in the box. The result suggests an ACC-CA1 interaction during observation that is specific to the observed, and thus the future, spatial trajectories.

## Interaction between ACC cells and CA1 place cells around SWRs

Since the replay of place cells within SWRs may be involved in planning future spatial trajectories (*Carr et al., 2011*; *Jadhav et al., 2012*; *Pfeiffer and Foster, 2013*; *Wu et al., 2017*), we asked whether ACC cells interacted with CA1 place cell activities associated with SWRs in the observation box. Because SWRs in the box primarily occurred during water consumption when individual place cells in CA1 ensembles fired spikes in population bursts, we examined firing activities of ACC cells, along with their CA1 ensembles, during water consumption periods in the box. As shown in *Figure 5A and B*, an example ACC right-selective cell and its CA1 same-side (right) ensemble tended to activate together with SWRs during the water consumption periods on the same side of their selectivity (right trials) in the box.

We quantified the firing rates of ACC cells, as well as their associated CA1 ensembles within SWRs, during the water consumption period of each trial in the box. Overall, both ACC cells and CA1 ensembles had higher firing rates on the same side of their selectivity than the opposite side under Demo, but not under Object or Empty (*Figure 5—figure supplement 1A–D*). However, this difference disappeared for ACC cells, but not for CA1 cells, around SWRs (*Figure 5—figure supplement 1E and F*), suggesting that the higher rate of ACC cells on the same side occurred during the broad water consumption period, not specifically within SWRs. We computed the correlation between the trial-by-trial firing rates of each ACC cell and those of its same- or opposite-side CA1 ensemble (same-side or opposite-side correlation). For the example ACC cell in *Figure 5A*, its same-side correlation was significant ($R=0.43$, p=0.0019, *Pearson's r*), but not its opposite-side one ($R=0.36$, p=0.0076), indicating that, when the cell displayed a high rate in the water consumption period of a trial (e.g. a right trial), the CA1 same-side (right) ensemble, but not the CA1 opposite-side (left) ensemble, was also activated with a high rate in the same trial (*Figure 5C*).

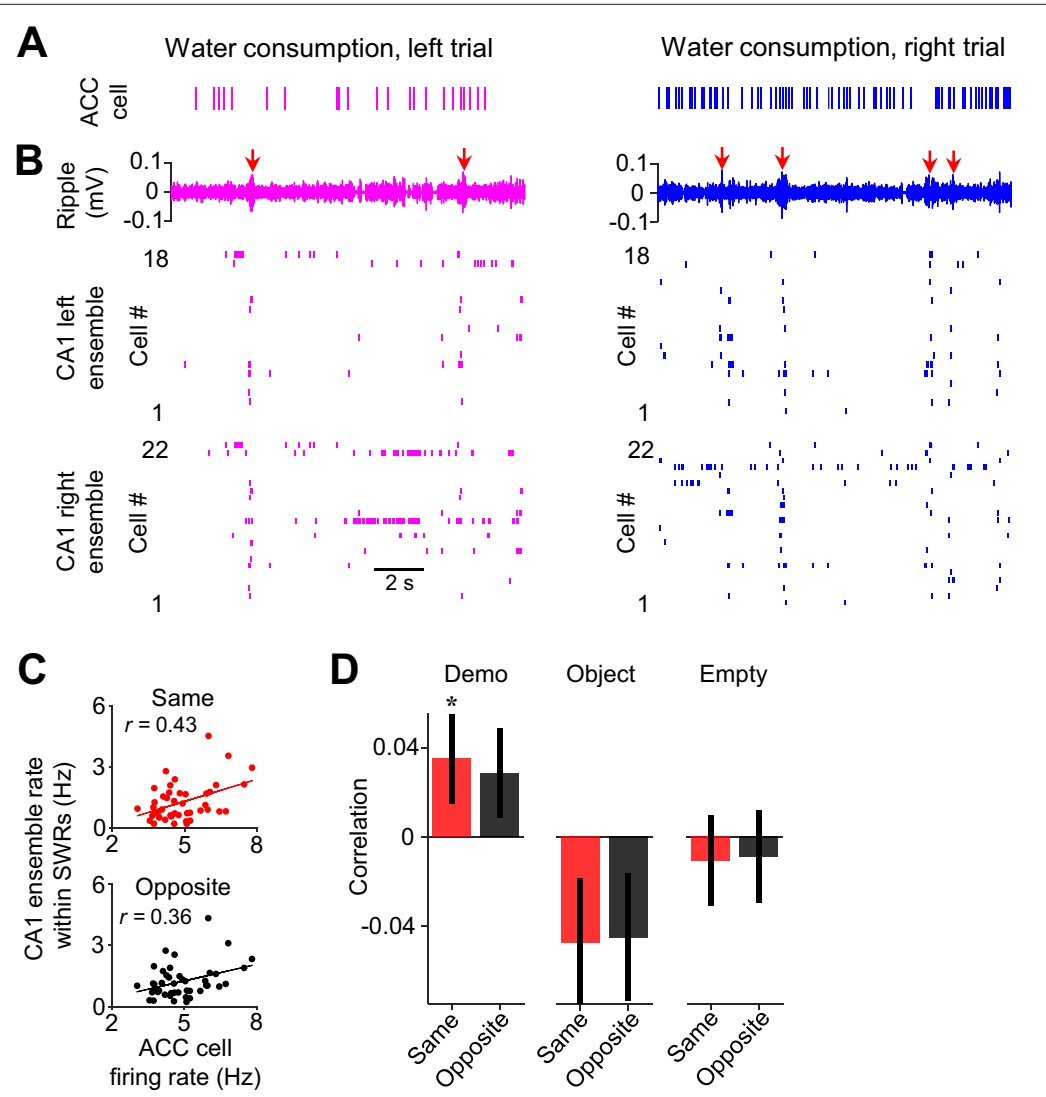

**Figure 5.** Activities of anterior cingulate cortex (ACC) cells were correlated with those of CA1 same-side ensembles during water consumption in the box. (**A**) Spike raster of an example ACC right-selective cell during water consumption in the box for a left and a right trial of a session. Each tick is a spike. (**B**) CA1 local field potentials (LFPs) filtered within the ripple band (top) and spike raster of CA1 cells in the left (middle) and right ensemble (bottom) during the same time periods in (**A**). Arrows: sharp-wave ripples (SWRs). Note the population bursts associated with SWRs. (**C**) Correlation (r) between trial-by-trial firing rates of an example ACC cell and its CA1 same- and opposite-side ensemble within SWRs during water consumption in the session. Each dot is a trial. (**D**) Average correlations (mean ± SEM) for all ACC selective cells and their CA1 same- and opposite-side ensembles, as computed in (**C**), under Demo, Object, and Empty.

The online version of this article includes the following figure supplement(s) for figure 5:

**Figure supplement 1.** Activities of anterior cingulate cortex (ACC) selective cells and CA1 ensembles during water consumption in the box differed between same and opposite sides of their selectivity.

Averaging over all ACC cells (**Figure 5D**), we found that the mean same-side correlation under Demo was significantly higher than 0 (0.035±0.019, N=202, p=0.04, one-sided t-test compared to 0). The mean opposite-side correlation showed a trend toward significance (0.028±0.019, N=202, p=0.08), likely due to a general interaction between CA1 activities around SWRs with prefrontal cortical areas including ACC, as shown in previous studies (**Jadhav et al., 2016**; **Remondes and Wilson, 2015**). However, the same-side correlation was significantly higher than the opposite-side one (p=0.047, one-sided *paired* t-test). In contrast, the same- and opposite-side correlations were not significantly higher

than 0 and not significantly different from each other under Object (same: –0.047±0.028, $N$=101, p=0.95, one-sided $t$-test compared to 0; opposite: –0.045±0.028, p=0.94, one-sided $t$-test compared to 0; p=0.68, one-sided *paired* $t$-test between same and opposite) or Empty (same: –0.011±0.020, $N$=130, p=0.70, one-sided $t$-test compared to 0; opposite: –0.0088±0.021, p=0.66, one-sided $t$-test compared to 0; p=0.65, one-sided *paired* $t$-test between same and opposite). The result suggests an interaction between ACC selective cells and SWR-associated CA1 activities, following their interaction during observation but prior to self-running, that is specific to the observed, and thus the subsequent future, spatial trajectories.

## ACC activities during observation are associated with specific spatial contents of CA1 place cell activities within SWRs

Given the specific ACC-CA1 interaction during delay periods and during water consumption afterward, we next asked how activities of ACC selective cells during observation of a trial were related to later CA1 activities within SWRs of the same trial.

We examined the firing activities of ACC cells during delay periods and their associated same- and opposite-side CA1 ensembles in SWRs during water consumption in the box (*Figure 6A–D*) and computed their firing rates in each trial. As shown in *Figure 6E*, the trial-by-trial firing rates in delay periods of an example ACC cell were correlated with the rates of its same-side CA1 ensemble in SWRs (same-side correlation: $R$=0.24, p=0.05, *Pearson's r*), but not with its opposite-side CA1 ensemble (opposite-side correlation: $R$=0.065, p=0.34), indicating that a high (low) firing rate of the ACC cell in a trial was associated with a high (low) rate of its CA1 same-side ensemble (not the opposite-side ensemble).

Averaging over all ACC cells (*Figure 6F*), we found that the mean same-side correlation under Demo in this case was significant (0.051±0.015, $N$=202, p=5.7 × 10$^{-4}$, two-sided $t$-test compared to 0). The mean opposite-side correlation again showed a trend toward significance (0.028±0.014, $N$=202, p=0.051), but was significantly lower than the same-side one (p=3.3 × 10$^{-10}$, two-sided *paired* $t$-test). In contrast, the same- and opposite-side correlations were not significant and not significantly different from each other under Object (same: –0.030±0.026, $N$=101, p=0.25, two-sided $t$-test compared to 0; opposite: –0.029±0.026, p=0.26, two-sided $t$-test compared to 0; p=0.87, two-sided *paired* $t$-test between same and opposite) or Empty (same: –0.004±0.019, $N$=130, p=0.83, two-sided $t$-test compared to 0; opposite: 0.0012±0.020, p=0.95, two-sided $t$-test compared to 0; p=0.35, two-sided *paired* $t$-test between same and opposite). In addition, we found a strong correlation between activities of ACC selective cells during observation and their activities during water consumption afterward in the box (*Figure 6—figure supplement 1*), suggesting a continuation of ACC cell activation in the box from observation to later water consumption of the same trial.

Finally, we analyzed how ACC activities during observation were related to the precise spatial contents encoded by CA1 replay activities within SWRs. For each of those SWRs identified as a replay event, we determined which of the maze trajectories was replayed, by Bayesian decoding using place cell templates during self-running (*Figure 7A*; *Davidson et al., 2009*; *Karlsson and Frank, 2009*; *Zhang et al., 1998*). We computed a correlation between the trial-by-trial firing rate of an ACC cell during delay periods and the number of replays for the inbound or outbound trajectory template on the same or opposite sides of its selectivity (same or opposite side, inbound or outbound template).

As shown in *Figure 7B*, the firing rate of an example ACC cell was significantly correlated with the number of replays for the same-side inbound template (same-side inbound correlation: $R$=0.45, p=0.0012, *Pearson's r*), but not for the opposite-side inbound one (opposite-side inbound correlation: $R$=–0.28, p=0.97), indicating that a high (low) firing rate of the ACC cell during observation in a trial was associated with a high (low) number of replays for its CA1 same-side, but not the opposite-side, inbound template.

Averaging over all ACC selective cells (*Figure 7C*), we found that the mean same-side inbound correlation under Demo was significantly higher than 0 (0.028±0.015, $N$=202, p=0.034, one-sided $t$-test compared to 0), but not the opposite-side inbound correlation (–0.012±0.015, $N$=202, p=0.78). The same-side inbound correlation was also significantly higher than the opposite-side one (p=0.048, one-sided *paired* $t$-test). Under the control condition of Object, the same- and opposite-side inbound correlations were not significantly higher than 0 and not significantly different from each other (same: –0.018±0.023, $N$=101, p=0.79, one-sided $t$-test compared to 0; opposite: –0.0024±0.023, p=0.54,

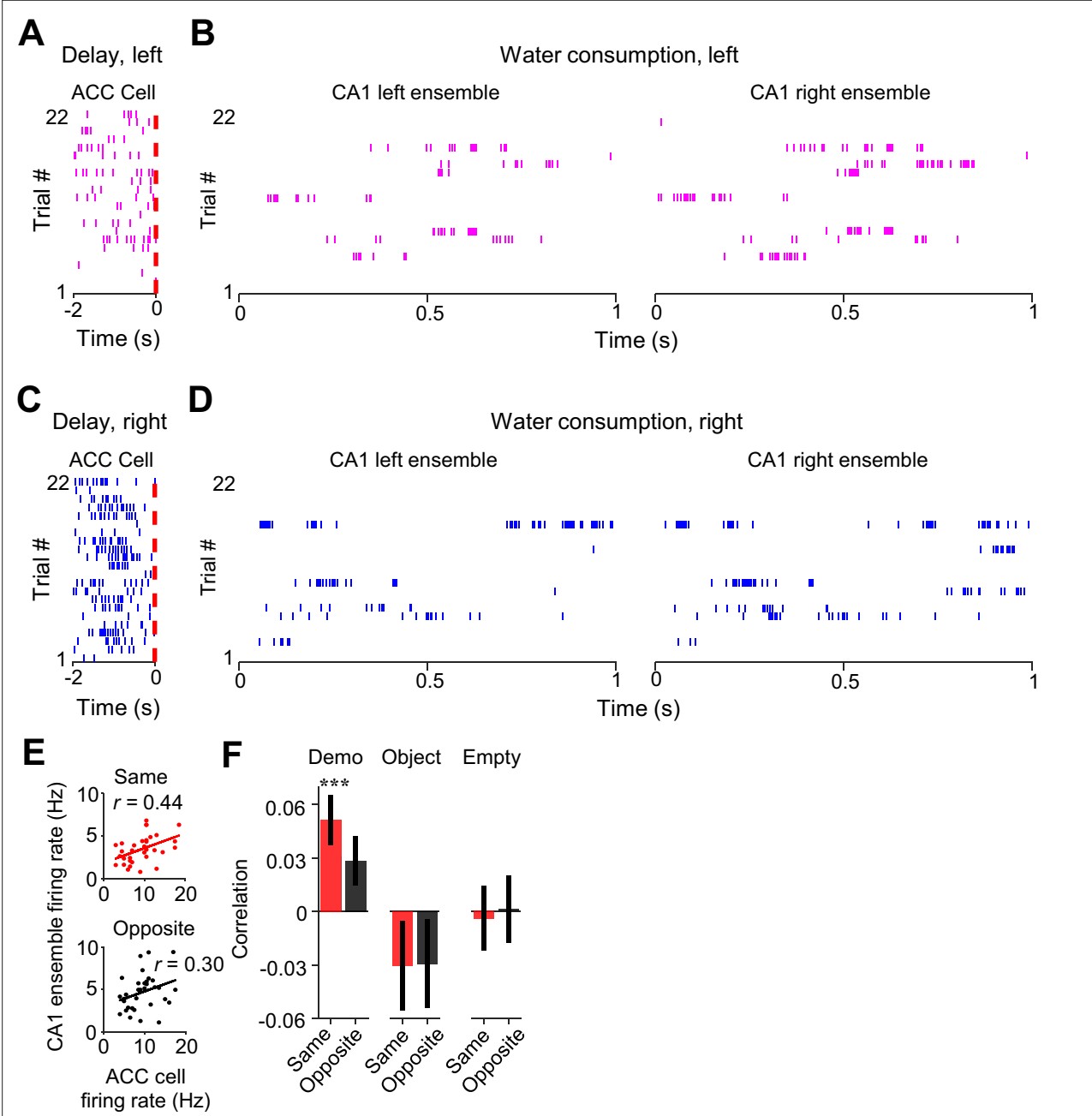

**Figure 6.** Activation of anterior cingulate cortex (ACC) selective cells during delay periods was associated with subsequent activation of CA1 same-side ensembles within sharp-wave ripples (SWRs) during water consumption in the box. (**A**) Spike raster of an ACC right-selective cell during delay periods of left trials in an example session. Each tick is a spike. Red dashed line: observer rat's (OB's) first rewarded poke time in the box (time 0). (**B**) Spike raster of CA1 left and right ensembles within SWRs during water consumption periods of the same (left) trials in (**A**). Only spikes within SWRs of a 1 s window are shown for clarity. (**C–D**) Similar to (**A–B**), but for the same ACC (right-selective) during delay periods and same CA1 ensembles during water consumption periods in the right trials of the same session. (**E**) Correlation (*r*) between trial-by-trial firing rates of the ACC cell during delay periods and its CA1 same- or opposite-side ensembles within SWRs during water consumption in the session. Each dot represents a trial. (**F**) Average correlations (mean ± SEM) for all ACC selective cells and their CA1 same- and opposite-side ensembles, as computed in (**E**), under Demo, Object, and Empty.

The online version of this article includes the following figure supplement(s) for figure 6:

**Figure supplement 1.** Activities of anterior cingulate cortex (ACC) selective cells during delay periods were correlated with their activities during water consumption periods in the box.

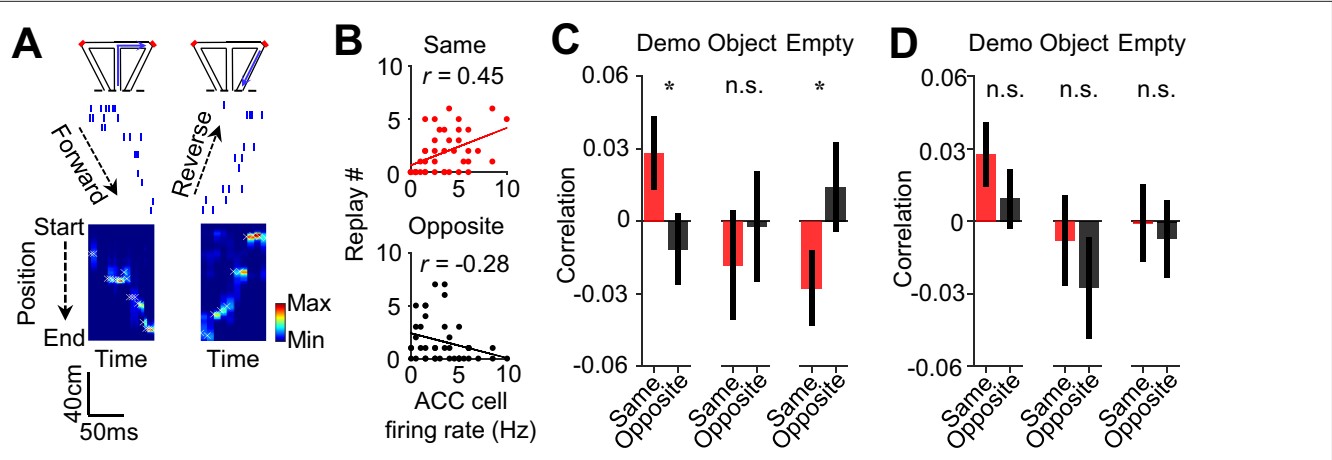

**Figure 7.** Activation of anterior cingulate cortex (ACC) selective cells during delay periods was associated with the replay of CA1 same-side ensembles during water consumption in the box. (**A**) Spike raster (middle) of an example forward/reverse replay of the CA1 template cells corresponding to the right outbound/inbound trajectory (top). The Bayesian-decoded probability of trajectory positions at each time bin is shown at the bottom. x: decoded (peak probability) positions. Dashed line: replay order of spikes. (**B**) Trial-by-trial correlation (*r*) between an example ACC cell's firing rate during delay periods and the number of replays for CA1 templates on the same- or opposite-side inbound trajectories of the ACC cell's selectivity. Each dot represents one trial. (**C**) Average correlation values (mean ± SEM) for all ACC selective cells and the corresponding replay numbers for same- or opposite-side templates, as computed in (**B**), under Demo, Object, and Empty. Note the higher same-side correlation under Demo, but not under Object or Empty. (**D**) Similar to (**C**), but for outbound trajectories.

one-sided *t*-test compared to 0; p=0.68, one-sided *paired t*-test between same and opposite). Under Empty, such same-side correlation was even significantly lower than 0 and lower than the opposite-side correlation (same: –0.028±0.016, *N*=130, p=0.044, one-sided *t*-test compared to 0; opposite: 0.014±0.019, p=0.23, one-sided *t*-test compared to 0; p=0.040, one-sided *paired t*-test between same and opposite). Thus, high ACC activities during observation were associated with high activities of same-side CA1 ensembles within SWRs only under Demo, but not under the two control conditions. In contrast, such association did not occur between ACC activities and opposite-side CA1 ensembles under all the conditions.

Interestingly, the difference in the ACC-CA1 association between the same side and opposite side was less evident for outbound templates (*Figure 7D*). The mean same-side outbound correlation under Demo was still significantly higher than 0 (0.027±0.013, *N*=202, p=0.020, one-sided *t*-test compared to 0), but not the opposite side (0.0092±0.012, *N*=202, p=0.23). However, the difference between the same- and opposite-side correlations did not reach a significance level (p=0.12, one-sided *paired t*-test). The same- and opposite-side outbound correlations were not significantly higher than 0 and not significantly different from each other under Object (same: –0.0082±0.019, *N*=101, p=0.67, one-sided *t*-test compared to 0; opposite: –0.028±0.021, p=0.90, one-sided *t*-test compared to 0; p=0.19, one-sided *paired t*-test between same and opposite) or Empty (same: –0.00082±0.016, *N*=130, p=0.52, one-sided *t*-test compared to 0; opposite: –0.0071±0.017, p=0.67, one-sided *t*-test compared to 0; p=0.39, one-sided *paired t*-test between same and opposite).

Taken together, our data support that activation of ACC cells selective to the observed trajectories during observation is associated with the later activation of CA1 ensembles in SWRs selective to the same trajectories. More specifically, such ACC activation during observation is associated with a bias of CA1 place cell replay toward the observed trajectories, especially inbound ones, later in the same trial in the box. Given that the replay for inbound trajectories predicts the trajectories during subsequent self-running in the maze (*Mou et al., 2022*), our result here suggests that ACC activities during observation in the box influence subsequent spatial decisions in the maze through their bias on CA1 replay in SWRs.

## Discussion

We have analyzed activities of ACC cells and their interactions with CA1 place cells in the oSWM task. We found that many ACC cells, selective to the left or right side of the T-maze during self-running, are also activated during observation (delay periods) with same-side selectivity in the observation box. Such ACC same-side selectivity is reduced in error trials. The activities of ACC selective cells are correlated with those of CA1 same-side ensembles during observation and during water consumption afterward. In addition, the ACC activities during observation are correlated with CA1 activities within SWRs and associated with a bias in the spatial contents of CA1 replay in the box toward future trajectories in the maze. These results reveal the selective activation patterns of ACC neurons during ongoing observation and how they interact with hippocampal place cells to guide subsequent spatial behavior.

The activation of ACC selective cells during observation supports a specific functional role of ACC in oSWM. ACC cells are selective to particular trajectories or segments of the maze during self-running, consistent with the relatively coarse spatial responses of prefrontal cortical neurons (compared to CA1 place fields) in previous studies (*Jadhav et al., 2016*; *Jones and Wilson, 2005*; *Mashhoori et al., 2018*; *Remondes and Wilson, 2015*; *Siapas et al., 2005*). It is unlikely that these ACC cells encode spatial locations similarly to place cells. Given that previous studies show that ACC activities are modulated by valence, value, or decision outcomes (*Caracheo et al., 2018*; *Hillman and Bilkey, 2010*; *Kane et al., 2022*; *Mashhoori et al., 2018*; *Monosov, 2017*), it is possible that the ACC selective activities in the maze reflect the reward or value associated with the spatial segments or trajectories being traveled. A major finding in this study is that these ACC cells, selective in the maze, are also selective to the same side of the observation box during delay periods. The oSWM task here utilizes a 'same-side' rule that the OB learns to run the same-side trajectories of the Demo in the maze, which by design is to include the two behavioral phases, observation and imitation, of the classical social learning theory (*Bandura, 1997*) in a single trial of the working memory task. The same-side selectivity of ACC cells indicates that, when the OB performs the task correctly in each trial, the ACC cells selective to the correct future trajectories during self-running are already activated during observation, but not those selective to the trajectories on the opposite side. Such same-side selectivity only occurs with the presence of the Demo, not under the control Object or Empty condition, and it is reduced in error trials, suggesting that the selective activation of ACC cells during observation is not simply due to the OB moving to the same side in the box. Our finding thus supports the following hypothesis: ACC neurons coding the value of trajectories in the maze during self-running are also activated during observation, cued by the observed Demo's action, to assign a 'vicarious value' to the correct future trajectories.

Our results also suggest that hippocampal place cells may play a functional role different from that of ACC cells in oSWM. CA1 place cells selective to the left or right side of the maze during self-running do not display same-side selectivity during observation in the box. Thus, the observation-induced neural activities relevant to spatial decisions appear to be more distinguishable in ACC neurons than in CA1 neurons. CA1 place cells may be more important in the water consumption period when SWRs occur. Evidence exists that CA1 activities within SWRs, especially the replay events, are involved in the planning of immediate future trajectories (*Carr et al., 2011*; *Jadhav et al., 2012*; *Pfeiffer and Foster, 2013*; *Wu et al., 2017*). Indeed, remote replay of inbound trajectories in the box can be used to predict the subsequent choice in the maze in this oSWM task (*Mou et al., 2022*). Therefore, it is possible that the CA1 place cell activities within SWRs in the box plan for the future trajectories to run in the maze.

How does the value information encoded in ACC cells during observation lead to a spatial plan later encoded in CA1 place cell activities within SWRs? First, our data show that the activities of ACC selective cells and their CA1 same-side ensembles are temporally correlated during delay periods, suggesting a functional interaction between ACC and CA1 cells during observation. Second, our data also show that the ACC activation during delay periods continues to the water consumption period in the box, when frequent SWRs occur. More importantly, same-side activities of ACC selective cells and their same-side CA1 ensembles are enhanced in a correlated fashion during water consumption in the box, and activities of CA1 same-side and opposite-side ensembles within SWRs become distinguishable. This ACC-CA1 correlation may not occur precisely within SWRs, but via a broader interaction during reward consumption, e.g., through an enhanced excitation from ACC to CA1. The interaction

can be mediated by the direct (*Cenquizca and Swanson, 2007*; *Rajasethupathy et al., 2015*) or indirect (*Bubb et al., 2017*; *Sakata et al., 2019*) anatomical connections between the hippocampus and medial prefrontal cortex, including ACC. It is believed that the occurrence of SWRs marks an internal hippocampal state for planning, memory recall, or consolidation (*Buzsáki, 2015*). Unlike the active state during observation when CA1 cells actively respond to input originating from sensory cortices, CA1 cells during SWRs may be driven more internally from prefrontal areas, including ACC (*Wang and Ikemoto, 2016*). Indeed, our data demonstrate that the activation of ACC selective cells during observation in a trial is accompanied by the preferential activation of their CA1 same-side ensembles and a bias of CA1 replay toward same-side trajectories prior to self-running. Taken together, our data are consistent with the following model: The continued ACC activation from delay periods to water consumption periods and a broad ACC-CA1 interaction enables the ACC cells selective to the future trajectories to bias the CA1 replay in the box, which in turn plans for subsequent self-running on the same trajectories in the maze.

In summary, our study of ACC cells and their interactions with hippocampal place cells in the oSWM task provides evidence that observing the action of a social subject activates specific ACC cells that encode value information involved in self-running decisions. These ACC cells interact with CA1 place cells for planning subsequent spatial behavior. ACC cells are known to be involved in observational learning (*Allsop et al., 2018*; *Burgos-Robles et al., 2019*; *Jeon et al., 2010*; *Olsson and Phelps, 2007*) and decision-making (*Hillman and Bilkey, 2010*; *Kane et al., 2022*). Our study thus provides a link between these properties of ACC cells and reveals a specific functional role for their interaction with CA1 place cells in observational learning of spatial navigation.

## Methods

### Animal

Sixteen 4–10 months' male Long-Evans rats (Charles River Laboratories), with a weight of 400–600 g, were used in this study. Among them, 10 rats were used in the lesion experiment and 6 were used for the electrophysiological recording experiment. All experimental procedures followed the guidelines from the US National Institute of Health and were approved by the Institutional Animal Care and Use Committee at Baylor College of Medicine (protocol AN-5234).

### Apparatus

Our behavioral apparatus consisted of a small observation box and a continuous T-maze with a resting box. Briefly, the trapezoidal observation box was placed ~30 cm away from the choice point of the T-maze. Both the observation box and the T-maze were elevated ~50 cm above the floor. There were two water ports mounted on two side walls of the observation box and two at the T-maze. An animal's nose poke at a water port triggered water delivery through peristaltic water pumps if correct behavioral response was performed.

### Behavioral procedure

Animals were water-deprived several days before the training started while food was available ad libitum at all times. Animals' weight was maintained at >85% of the ad libitum level. Every rat in this study was first trained as an OB in the oSWM task to reach a criterion performance (>70% correct trials in the maze for at least 2 consecutive days). The electrophysiological recording or behavioral testing in the lesion experiment was then performed in daily sessions of ~90 min, each with a well-trained OB performing the oSWM task for ~40 trials.

Three behavioral conditions were used in the electrophysiology experiment. (1) *Demo*. This was the standard experimental condition. In each trial of the task, a well-trained demonstrator (Demo) randomly chose the left or right side (outbound trajectory) and then poked for water reward in the T-maze, while a well-trained OB observed the Demo and was rewarded with water if the OB poked on the same side in the observation box as the Demo's in the maze within 10 s of the Demo's first poke. The Demo returned via a left or right sidearm (inbound trajectory) and was confined in a rest box. The OB was transported to the T-maze at the beginning of the central arm and rewarded with water if the animal ran to the same side of the maze (outbound trajectory) as the Demo did (same-side rule). The OB returned via the same sidearm (inbound trajectory) as the Demo did and was transported back to

the observation box before the next trial started. An error trial in this case was when the OB ran to the opposite side of the maze to the Demo's and was not rewarded in the maze. (2) *Object*. In each trial under this condition, the Demo was replaced by a moving object (10 cm × 20 cm black rectangle plastic block attached to the end of a 1.5 m wood pole). The movement of the object mimicked the Demo's movement in the T-maze. The moving object triggered the water ports and then remained at the reward location for a similar duration as the Demo rat did under the Demo condition. A trial was considered correct (an error) if the OB's choice in the maze was the same (or opposite-side) water port the object activated. (3) *Empty*. In each trial under this condition, the Demo was removed and the T-maze was left empty. A manually controlled wood pole triggered the water ports every ~2 min and remained at the reward location for a similar duration as the Demo rat did under the Demo condition. A trial was considered correct (or an error) if the OB chose the same (or opposite-side) water port activated manually.

Only one condition was conducted per day. When multiple conditions were used for the same OB, the OB was re-trained before the next condition under the standard Demo condition if the OB's performance in the maze was below the criterion under the previous condition. Specifically, one rat underwent six Demo sessions, then two Empty sessions; one rat underwent three Demo sessions, then two Empty sessions; one rat underwent five Demo sessions, then two Object sessions, then two Empty sessions; one rat underwent two Demo sessions, then four Object sessions, then three Empty sessions; one rat underwent five Object sessions, then five Empty sessions; one rat underwent three Demo sessions only.

## Surgery for electrophysiological recording

Six well-trained OBs were surgically implanted with a hyperdrive consisting of 22 independently movable tetrodes and two reference electrodes, similarly to previous studies (*Haggerty and Ji, 2015*; *Mou et al., 2022*). The electrodes targeted two regions: the right dorsal ACC at coordinates anteroposterior (AP) 1.6 mm and mediolateral (ML) 1.0 mm from the Bregma, and the right dorsal hippocampal CA1 region at coordinates AP –3.8 mm, ML 2.4 mm. Animals were anesthetized using isoflurane (0.5–3%), and the hyperdrive was fixed to the rat skull through dental cement and anchoring screws.

## Recording procedure

Within 2–3 weeks following the surgery, hippocampal tetrodes were slowly advanced to the CA1 pyramidal layer until characteristic SWRs were observed (*Buzsáki et al., 1992*). ACC tetrodes were left in the cortex. One tetrode placed in the white matter above the CA1 and one silent tetrode in ACC were selected as the reference tetrode for CA1 and ACC, respectively. Recording started only after the tetrodes had not been moved for at least 24 hr.

Within about 2–3 weeks after the surgery, the OBs were accustomed to the hyperdrive and re-trained back to the criterion performance under the standard Demo condition (under Demo). Afterward, each OB was recorded while performing the oSWM task for 5–11 consecutive daily sessions under Demo, Object, or Empty, as described above.

## Lesion experiment

Ten well-trained OBs were used for the lesion experiment to examine whether the oSWM task was ACC-dependent. Rats were randomly assigned to a lesion (*N*=5) or a control group (*N*=5). For the lesion group, neurotoxic lesions in ACC were made by infusing NMDA (Sigma-Aldrich, St. Louis, MO, USA) in a phosphate-buffered saline (PBS) vehicle (20 μg/μL made by 100 mM PBS, pH = 7.4), bilaterally at a rate of 0.2 μL/min to each of three ACC sites per hemisphere using a microinfusion pump (KD Scientific, Holliston, MA, USA) and a 10 μL Hamilton syringe (Hamilton, Reno, NV, USA). The coordinates of the infusion sites were: (AP 2.1 mm, ±ML 0.6 mm, –2.5 mm ventral to the dura [DV]), (AP 1.2 mm, ±ML 0.6 mm, DV –2.5 mm), (AP 0.3 mm, ±ML 0.6 mm, DV –2.5 mm). A total of 0.2 μL was injected at each site, and the syringe was left at each infusion site for 3 min. For the control group, 0.2 μL PBS vehicle alone was similarly infused to each of the six coordinates. After the animals fully recovered from the surgery (~14 days), they were tested in the oSWM task under the standard Demo condition.

## Histology

To verify the recording and infusion sites, all OBs in the recording and lesion experiments were euthanized (pentobarbital, 150 mg/kg) and subjected to histology. For the recorded animals, a small lesion at each recording site was generated by passing a 30 μA current for 10 s on each tetrode. Brain tissues were fixed (10% formaldehyde solution overnight) and sectioned at 90 μm thickness. Brain sections were stained using 0.2% Cresyl violet and coverslipped for storage. Tetrode recording locations were identified by matching the lesion sites with tetrode depths and their relative positions.

## Data acquisition

A Digital Lynx acquisition system (Neuralynx, Bozeman, MT, USA) was used to record spikes and LFP signals, as described previously (*Haggerty and Ji, 2015*; *Mou and Ji, 2016*; *Wu et al., 2017*). A 600 μV threshold was set for spike detection. Spike signals above this threshold were sampled at 32 kHz and digitally filtered between 600 Hz and 9 kHz. LFP signals were sampled at 2 kHz and filtered between 0.1 Hz and 1 kHz. The head and body center positions of each animal were tracked at 30 Hz with a resolution of approximately 0.1 cm by the EthoVision XT system (Noldus, Leesburg, VA, USA).

## Behavioral quantification

We quantified each OB's behavioral performances in the observation box and in the T-maze. Briefly, the performance in the box was quantified by a poke performance curve for each session, which was the normalized poke rate at each 0.25 s time bin in the box, aligned relatively to the corresponding Demo's poking time in the maze in each trial (time 0). The performance in the maze was quantified by the percentage of correct trials for each session.

## Cell inclusion

Spikes detected using Neuralynx were sorted into single units (single cells) off-line, using a custom software (xclust, M Wilson at MIT, available at GitHub: https://github.com/wilsonlab/mwsoft64/tree/master/src/xclust; *Layton, 2020*). Some cells might be repeatedly sampled across sessions since we did not track cell identities across multiple recording sessions. A total of 1049 ACC cells and 2102 CA1 cells were obtained in 44 sessions from 6 OB rats. Among the CA1 cells, 1226 were classified as putative CA1 pyramidal neurons active (mean firing rate between 0.4 and 10 Hz) in the T-maze. These active CA1 cells and all ACC cells were included in the analysis of neural data.

## ACC selective cell, SI, and SI correlation

For each ACC cell, we computed its firing rate on the left or right side of the maze during self-running for each trial, by dividing the number of spikes occurring during active running, with the stopping periods (velocity <5 cm/s for >3 s) excluded, on the outbound and inbound trajectory combined for each side over the total amount of running time. A cell was classified as a selective cell if its mean rates on the left and on the right side of the maze averaged over all correct trials were significantly different (p<0.05, two-sided *t*-test).

Further analysis on ACC was performed only on ACC selective cells unless otherwise specified. For each such ACC selective cell, we also computed the firing rates during the delay period of each trial in the observation box, defined as the 2 s before the first correct (rewarded) poke in the box, and considered it selective in the box if its mean rates during delay periods between left and right correct trials were significantly different. A cell was determined to have same- or opposite-side selectivity if it was selective in the box to the same side as or different side from that in the maze.

To quantify the significance of the percentage of ACC cells with same- or opposite-side selectivity (among all ACC selective cells), we randomized every cell's rates during delay periods of left or right trials and recounted the percentage of cells with same- or opposite-side selectivity. This was repeated 1000 times, and a (chance-level) distribution of the resulting 1000 percentage numbers was obtained. The actual number was z-scored relative to the chance-level distribution with its significance determined by the Z-test.

For every cell, we also computed an SI, given its mean rates on the left ($FR_{left}$) or right ($FR_{right}$) side in the maze or box, as

$$SI = \frac{FR_{right} - FR_{left}}{FR_{right} + FR_{left}}. \tag{1}$$

To assess the level of same-side selectivity at the population level, we correlated the SIs in the box for all ACC selective cells and their SIs in the maze, with the significance determined by *Pearson's* correlation coefficient *r*. The same analysis was applied to individual CA1 cells active in the maze to assess their same-side selectivity during delay periods in the box.

## CA1 SWR detection

SWRs were detected as population burst events (PBEs) from the CA1 multiunit activity (MUA), as in previous studies (*Diba and Buzsáki, 2007*; *Mou et al., 2022*; *Wu et al., 2017*). In each session, all putative spikes recorded by all CA1 tetrodes were binned in 10 ms time bins. Spike counts in each bin were counted and smoothed by a Gaussian kernel with a $\sigma$ of two bins and standardized from 0 to 1. A PBE was defined as a time period such that the standardized peak MUA spike counts in this period exceeded a threshold of 0.35, and the MUA spike counts at the start and end times exceeded a threshold of 0.15. Adjacent PBEs with a gap <30 ms were combined.

## CA1 place cell ensembles and replay detection

For each CA1 cell active during self-running in the T-maze, we constructed its firing rate curves on the four maze trajectories (left/right, inbound/outbound). For each trajectory of a session, we then constructed a place cell template. A CA1 cell was eligible to be included in a template if its rate curve on the trajectory had a peak firing rate of at least three standard deviations above its mean firing rate. A CA1 left or right ensemble was made of all cells in the inbound and outbound templates combined on the left or right side of the T-maze. The firing rate of an ensemble was computed by aggregating all spikes from all cells in the ensemble within a time bin.

We identified whether CA1 place cell activities within an SWR (PBE) replayed a place cell template on a trajectory by a Bayesian decoding method, as described in previous studies (*Davidson et al., 2009*; *Karlsson and Frank, 2009*; *Mou et al., 2022*; *Wu et al., 2017*; *Zhang et al., 1998*). Briefly, PBEs with at least four active CA1 template cells were defined as candidate PBEs for the template. For each candidate PBE, we assumed independent *Poisson* processes among the cells in the template and computed the posterior spatial probability distribution for each time bin (20 ms with a 10 ms overlapping step). The decoded position at each time bin was the location on the template trajectory with the maximum posterior probability. We then correlated the decoded positions and the time bin numbers. The *Pearson's* correlation coefficient *r* was compared to 1000 shuffle-generated coefficients, obtained by correlating the decoded positions with randomly shuffled time bin numbers 1000 times. The proportion of shuffle-generated coefficient values greater than the actual one gave the significance p-value. A candidate PBE was considered a replay if p<0.05.

## Firing rate difference in error trials and during water consumption

For each ACC cell with same-side selectivity, we computed its firing rates during delay periods on its preferred (left or right) side in correct and error trials of a session, if the session contained at least three error trials on its preferred side (the OB rewarded on its preferred side in the box, but ran incorrectly to the opposite side in the maze). A firing rate DI was defined similarly as in *Equation 1* above. The rate difference between correct and error trials was assessed at the population level by comparing the mean DI among all ACC cells with same-side selectivity to 0 by two-sided *t*-test. The same analysis was also applied to ACC cells with same-side selectivity in error trials occurring on their non-preferred side (the OB rewarded on the non-preferred side of a cell in the box, but ran incorrectly to the opposite side in the maze).

For each ACC cell, we also compared its firing rates on the same side or opposite sides of its selectivity during water consumption periods of a session, similarly by a firing rate DI computed as in *Equation 1*. The same was performed on CA1 activities during water consumption. Since CA1 firing activities within and outside SWRs are considered to reflect different behavioral states (*Buzsáki, 2015*; *Buzsáki et al., 1992*; *Csicsvari et al., 2000*), we restricted our analysis to CA1 activities during water consumption within SWRs. Since each individual CA1 cell only fired very few or no spikes, we combined all spikes of cells in a CA1 ensemble within an SWR to obtain an aggregated firing rate.

The difference between mean rates within the SWRs occurring on the same and opposite sides of an ensemble in the observation box (left or right for CA1 left or right ensembles) was quantified by a firing rate DI.

## Firing rate correlation

To quantify ACC-CA1 interactions, we computed firing rate correlations between ACC selective cells and their CA1 same- or opposite-side ensembles in the observation box. During delay periods, we computed firing rates of each ACC selective cell in 200 ms time bins. Given the relatively sparse firing of CA1 cells, to avoid the bias of correlation by firing rate, we computed the rate of a CA1 ensemble by counting all the aggregated spikes of all cells in the ensemble within each time bin. The binned firing rates of an ACC cell and its CA1 same- or opposite-side ensemble of all delay periods in a session were correlated by *Pearson's* correlation coefficient to quantify the temporal correlation between the ACC cell and the CA1 ensemble selective to the same or opposite sides of its selectivity.

During water consumption, we isolated spikes of CA1 ensembles within SWRs, for the same reason that spikes within and outside SWRs reflect different behavioral states, and computed their rates in each trial. We also computed the rate of each ACC selective cell during the entire water consumption period in each trial, without considering specific time periods since no specific activity patterns like CA1 population bursts in SWRs were observed. For all trials in a session, the trial-by-trial firing rates of an ACC cell were correlated with those of its CA1 same or opposite side by *Pearson's r*. A significant, high value of such trial-by-trial correlation means that, when an ACC cell selective to the left or right side of the maze fired with a high (or low) firing rate during the water consumption period of a trial, the CA1 ensemble on the same side of the maze was also activated with a high (or low) rate within SWRs in the same period of the same trial.

Similarly, this trial-by-trial correlation was also used to quantify the relation between the activity of an ACC selective cell during the delay period of a trial and its rate during the water consumption or the rate of its CA1 same/opposite-side ensemble within SWRs during the water consumption of the same trial. In addition, the trial-by-trial firing rates of an ACC cell during delay periods were also correlated with the trial-by-trial numbers of replay for the inbound or outbound trajectory template during water consumption.

## Statistical analysis

Sample sizes were decided based on standards in the field. No sample-size calculations were performed prior to experimentation. Criteria for excluding individual cells or sessions from analyses were detailed above. Animals (OBs) were not allocated into experimental conditions completely randomly since all animals need to be trained with a Demo. Therefore, the Demo condition was always tested first before animals were assigned to other behavioral conditions. Investigators and animals were not blind to the condition allocation during data collection because the setup configurations and experimental conditions were evident to both investigators and animals by design. Analysis tools and codes were automated and applied to different conditions uniformly; therefore, blinding was not relevant.

All statistical analyses were performed using MATLAB functions. The statistical significance level was set at $p < 0.05$. Some specific statistical tests are described above. In general, for parametric analyses, values are reported as mean ± SEM. *Two-way ANOVA* was used for multiple comparisons followed by post hoc *Fisher's* least significant difference test corrected by multiple comparisons. *A paired t*-test *or t*-test was used to compare between two groups or between a sample of values and a fixed value (e.g. 0). One-sided or two-sided comparison was specified wherever used. For nonparametric analyses, values are reported as median (25th percentile 75th percentile). The *Kruskal-Wallis* test followed by Dunn's test was used for multiple comparisons. The *Wilcoxon rank-sum* test or *signed-rank* test was used to compare between two groups. All statistical methods and sample size (*N*) were detailed in the main text or figure legends wherever used.

## Acknowledgements

We thank Ji lab members for discussions. We thank the J Chin lab for suggestions on lesion experiments. This work is supported by grants from the National Institute of Mental Health (R01MH106552, R01MH112523, and R01MH134897 to DJ).

## Additional information

### Funding

| Funder | Grant reference number | Author |
|---|---|---|
| National Institute of Mental Health | R01MH106552 | Daoyun Ji |
| National Institute of Mental Health | R01MH112523 | Daoyun Ji |
| National Institute of Mental Health | R01MH134897 | Daoyun Ji |

The funders had no role in study design, data collection and interpretation, or the decision to submit the work for publication.

### Author contributions

Xiang Mou, Conceptualization, Resources, Data curation, Software, Formal analysis, Supervision, Validation, Investigation, Visualization, Methodology, Writing – original draft, Project administration, Writing – review and editing; Daoyun Ji, Conceptualization, Resources, Data curation, Software, Formal analysis, Supervision, Funding acquisition, Validation, Investigation, Visualization, Methodology, Writing – original draft, Project administration, Writing – review and editing

### Author ORCIDs

Xiang Mou ⓘ https://orcid.org/0000-0002-8579-7316
Daoyun Ji ⓘ https://orcid.org/0000-0003-4115-5888

### Ethics

All experimental procedures in this study followed the guidelines from the US National Institute of Health and were approved by the Institutional Animal Care and Use Committee at Baylor College of Medicine (protocol AN-5234). All surgery was performed under sodium pentobarbital anesthesia, and every effort was made to minimize suffering.

Reviewer #1 (Public review): https://doi.org/10.7554/eLife.97884.3.sa1
Reviewer #2 (Public review): https://doi.org/10.7554/eLife.97884.3.sa2
Reviewer #3 (Public review): https://doi.org/10.7554/eLife.97884.3.sa3
Author response https://doi.org/10.7554/eLife.97884.3.sa4

## Additional files

### Supplementary files
MDAR checklist

### Data availability

All behavioral, electrophysiological data have been deposited at Dryad (https://doi.org/10.5061/dryad.dncjsxmc6). Any additional information required to reanalyze the data reported in this study is available from the lead contact upon request.

The following dataset was generated:

| Author(s) | Year | Dataset title | Dataset URL | Database and Identifier |
|---|---|---|---|---|
| Xiang M, Daoyun J | 2025 | Observational activation of anterior cingulate cortical neurons coordinates hippocampal replay in social learning | https://doi.org/10.5061/dryad.dncjsxmc6 | Dryad Digital Repository, 10.5061/dryad.dncjsxmc6 |

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
