## [Editor Report · eLife Assessment]

This study provides **convincing** evidence of coordinated spiking activity of neurons in the anterior cingulate cortex (ACC), and correlated activity in the CA1 subregion of the hippocampus, during observational learning. The authors also show coordinated ACC-CA1 neural activity during rest periods prior to the performance of the observationally learned task. The **important** findings significantly advance the field's understanding of neural mechanisms underlying social learning.

---

## [Referee Report · Reviewer #1 (Public review)]

Summary:

Mou and Ji investigate the relationship between firing rates in the anterior cingulate cortex (ACC) and CA1 neurons during observational learning. They found trajectory-selective responses in the ACC, coordinated activity between ACC and CA1 place cells for specific trajectories, and reactivation of these ensembles during sharp-wave ripples (SWRs), particularly during hippocampal replay events. The study is methodologically sound, the data are clearly presented, and the conclusions are well supported. The work is both novel and highly relevant to our understanding of social learning. Compared to the previous version of the paper, they have added substantial characterization of neuronal properties related to their firing during the task and replay events. I believe that the authors have therefore addressed most of my concerns and recommend the paper for publication as is.

Strengths:

The study is well designed, the data presented is very clear and the conclusions are appropriate regarding their results. The study is novel and of high relevance for the understanding of social learning.

Weaknesses:

All previous weaknesses have been addressed.

---

## [Referee Report · Reviewer #2 (Public review)]

Summary:

In the manuscript, Xiang Mou and Daoyun JI investigate how ACC neurons activated by observational learning communicate with the hippocampus. They assess this line of communication through a complex behavioral technique, in vivo electrophysiology, pharmacological approaches, and data analytical techniques. Firstly, authors find that observational performance is dependent on the ACC, and that the ACC possess neurons that show side selectivity (trajectory related) in both the observation box, when shuttling to reward, and during subsequent maze running, shuttling to the corresponding same side for reward. The side-selective activation appears stronger for correct trials compared to error trials specifically during observation of Demo rats. They compare how the CA1 of the hippocampus encodes these two environments and find that ACC side-selective neurons show correlation with side-selective CA1 ensembles during maze behavior, water consumption, and sharp-wave ripples.

Strengths:

Overall, the paper provides strong evidence that ACC neurons are activated by observational learning and that this activation seems to be correlated with CA1 activity.

Weaknesses:

Concerns, however, surround the strength of evidence that links ACC and CA1 activity during observational learning. Only weak correlations between the two regions are shown, and it is unclear if the ACC may lead CA1 activity or vice versa. It is possible that these processes reflect two parallel pathways. Without manipulation of ACC, it is difficult to assess whether ACC activity influences hippocampal replay.

Comments on revisions:

Lines 361-362: R and P values do not match that of Figure 5C.

---

## [Referee Report · Reviewer #3 (Public review)]

Summary:

Mou and Ji investigated neuro-computational mechanisms behind observational spatial learning in rats and reported several signs of functional coupling between the ACC and CA1 at the single neuron level. Using multi-site tetrode recording, they found that ACC cells encoding a path in a maze were activated while a rat observed another rat taking that path. This activation was also correlated with the activation of CA1 cells encoding the same path and facilitated their replay during sharp-wave ripples (SWRs) before the recording rat ran on the maze by itself. These activity patterns were associated with correct path choice during self-running and were absent in control conditions where the recording rat did not learn the correct choice through observations. Based on these findings, the authors argue that ACC cells capture the critical information during observation to organize hippocampal cell activity for subsequent spatial decisions.

Strengths:

The authors used multiple outcome measures to build a strong case for path-specific spike coordination between ACC and CA1 cells. The analyses were conducted carefully, and proper control measures were used to establish the statistical significance of the detected effects. The authors also demonstrated tight correlations between the spike coordination patterns and the successful use of observed information for future decisions.

Weaknesses:

(1) As evidence for the activation of path information in the ACC during observation, the authors showed positive correlations between firing rates during observation and those during self-running. This argument will be solidified if the authors use a decoding approach to demonstrate the activation of path-selective ACC ensemble activity patterns during observation. This approach will also open the door to uncovering how the activation of ACC path representation is related to the moment-to-moment position of the demonstrator rat and whether it is coupled with the timing of SWRs.

(2) The authors argued that the ACC biases the content of awake replay in CA1 during SWRs in the observation period. The reviewer wonders if a similar bias also occurs during SWRs in the self-run period (i.e., water consumption after the correct choice). This analysis will help test whether the biased replay occurs due to the need to convert observed information into future choices.

(3) Although the authors demonstrated the necessity of the ACC for the task, it remains to be determined whether firing coordination between the ACC and CA1 during observation is necessary for the correct path choice during self-runs. Some discussion on this point, along with future direction, would be beneficial for readers.

Comments on revisions:

The authors fully addressed my recommendations. I do not have any further comments.

---

## [Author Response]

The following is the authors’ response to the original reviews.

**Reviewer #1 (Recommendations For The Authors):**
Minor:(1) In Figure 2, only the right or left selective neurons are presented for the comparison, it would be helpful to also compare these with the neurons that are not selective for any of the sides and maybe include them in the supplemental materials

We have included all non-selective neurons in Figure 2D and supplemental Figure 2B. Their differences in firing rate between left and right sides are quantified by their selective indices (SIs).

(2) The authors should provide controls of speed during NMDA infusion and vehicle.

We have quantified and compared the duration of running laps, which is equivalent to speed.

(3) In Figure 1d, the trend shows that even during NMDA infusion, the animals learn as shown by a higher proportion of correct trials in the 3rd compared to the 1st trial

We thank the reviewer for pointing that out. We noticed that NMDAlesioned ACC animal showed a trend of improved performance in the track, and we believe this is due to re-learning of the task, which we point out in the main text. However, we emphasize that, compared to the Vehicle control, the overall performance of NMDA-lesioned animals was significantly impaired.

(4) Clarify the implications of the NMDA experiments, as it is not straightforward to interpret that an interplay between ACC-CA1 is involved in this task as per this experiment.

Rather than stating the involvement of ACC-CA1 interplay, we use the results of NMDA lesion experiment to demonstrate that ACC is also required, besides CA1, for the task.

(5) In Figure 4b, there seems to be a lag between CA1 and ACC correlations; the authors could provide a quantification of this temporal delay between CA1 and ACC.

Figure 4B shows the cross-correlation between one example ACC cell and its associated CA1 ensembles on the left and opposite sides. There was a broad peak around time lag 0. Our further investigation did not identify a significant, systemic delay for all ACC cells, which led us to quantify the correlation at time lag 0 in Figure 4C and D.

(6) The example correlation provided in 5c for the opposite, doesn't seem representative of the population trend as shown in 5d, since both the Same and the Opposite for the demo show a positive trend. It would be best to choose an example that represents the population better.

Following the reviewer’s suggestion, we have replaced the original plot with another ACC cell in Figure 5C.

(7) Almost the same can be applied to Figure 6.

Following the reviewer’s suggestion, we have replaced the original plot with another ACC cell in Figure 6E.

(8) The results in Figure 7 are convincing, in my opinion, as they show that the trend is lost for the opposite side (contrary to the coactivation shown in Figures 5 and 6 that showed the same trends for the same and opposite during Demo). Do the authors have any interpretation of this? Is it due to co-activity reflecting other task-relevant features different than the spatial trajectory being observed?

The correlation on the opposite side between CA1 and ACC shown in Figure 5C-D and Figure 6E-F is likely due to a general interaction between CA1 activities around SWRs with prefrontal cortical areas including ACC, as shown in previous studies (Jadhav et al., 2016; Remondes and Wilson, 2015). We would like to point out that this correlation only quantifies the coactivation between CA1 ensemble firing rates and individual ACC cells’ firing rate. This raw correlation does not consider the content of spikes generated by CA1 ensembles, neglecting the sequential firing patterns of CA1 cells. The replay analysis in Fig. 7 examines the order of spikes generated by individual CA1 cells. The result in Fig. 7 shows that the sequential activation of CA1 place cells more accurately reflects the distinction between the same- and opposite-side trajectories. We consider Fig. 7 is more refined analysis than Figs. 5 and 6.

(9) For all the figures regarding SWR activities, the authors should provide average PSTH for CA1 as well as ACC, perhaps also examples of neurons that are selectively active during one side or the opposite side runs.

Following the reviewer’s suggestion, we have added data to show PSTH for CA1 and ACC cells surrounding SWR peaks (Figure S5E, F).

**Reviewer #2 (Recommendations For The Authors):**
Below are additional notes for improvements.(1) Figure 1C. Unclear what Time 0 indicates.

We specify it (OB's poke time) in the figure legend.

(2) Figure 2C. Unclear what the numbers above datapoints mean.

Those numbers are selection indices (SIs), as specified in the legend.

(3) Figure 5: Line 374-375. Given the repetitive nature of the task, it is unclear whether SWRs are encoding upcoming or past spatial trajectories or whether they are encoding trajectories at all. The authors would need to show that SWRs-ACC communication is predictive of task outcome to claim it is specifically necessary for future outcomes rather than consolidating past trajectories.

We agree with the reviewer and have made changes to reflect that the ACC-CA1 correlation in Fig.5 is specific to the same side of their selectivity, not exactly to future trajectories. Regarding the repetitive nature of the task (same-side rule), we have specifically addressed the advantage and limitation of this task design in the discussion. Regarding the observer's own past vs. future trajectories, our past publication (Mou et al., 2022) shows that the CA1 replay in SWRs more likely encode the correct, future trajectories.

(4) Figure 7. It appears that the correlation was conducted between ACC activity and CA1 replays recorded at distinct time windows (delay period vs. water consumption). It is unclear how ACC activity could influence CA1 replays when they occur hundreds of milliseconds apart or even longer.

We thank the reviewer for raising this important question. We have shown that the higher same-side ACC activity during observation continues during water consumption. However, our added data in Fig.S5E show that this enhancement did not occur precisely within SWRs. We thus propose a possibility that the overall enhanced activity of same-side ACC cells during water consumption provides an overall, background excitation boost to same-side CA1 cells to enhance their replay within SWRs. We have revised the discussion section to present this model.

(5) Abstract: lines 24-25 Discussion: lines 475-476 Based on the data there is no certainty whether ACC biases or coordinates CA1 replays. The data simply shows that they are correlated with one another.

We have modified those sentences to clarify the non-causal nature of the interaction.

**Reviewer #3 (Recommendations For The Authors):**
Please see below for the list of minor corrections and suggestions:(1) Line 136-143: On the data shown in Figure 1D, I recommend using two-way mixed ANOVA with sessions as a within-subjects factor and groups as a between-subjects factor.

We thank the reviewer for this point. We indeed use two-way ANOVA for those comparisons. We have specified out in the text.

(2) Line 219-228: I recommend expanding the explanation of two control conditions here. It was written in the method section, but the readers would appreciate the gist of these conditions in the result section. In particular, it was unclear how box SI was calculated in the Empty condition. Also, the plots of poke rates in the control conditions will be useful to show that rats did not learn the correct choice from observation in these control conditions.

We have added more explanation of the two control conditions in the text. The quantifications of poke rates for Demo and two control conditions (Object, Empty) are provided in our previous publication (Mou et al., 2022).

(3) Line 610: Please specify the number of three types of sessions each rat underwent and the order of these session types.

We revise the texts in the Method section and provide the numbers.

(4) In Figure 2c legend, please specify what the number (e.g., -0.41) indicates.

Those numbers are selection indices (SIs), as specified in the legend.